# The Diagnostic Yield of Cerebrospinal Fluid Analysis for the Diagnosis of Primary Central Nervous System Lymphoma: A Systematic Review

**DOI:** 10.3390/cancers17142352

**Published:** 2025-07-15

**Authors:** Josephus L. M. van Rooij, Tom J. Snijders, Prerana Bhande, Tatjana Seute, Monique C. Minnema, Peter H. Wessels

**Affiliations:** 1Department of Neurology and Neurosurgery, UMC Utrecht Brain Center, University Medical Center Utrecht, 3584 CX Utrecht, The Netherlands; j.l.m.vanrooij-7@umcutrecht.nl (J.L.M.v.R.); t.j.snijders@umcutrecht.nl (T.J.S.); prerana.b.pai@gmail.com (P.B.); t.seute@umcutrecht.nl (T.S.); 2Department of Neurology, St. Antonius Hospital, 3543 AZ Utrecht, The Netherlands; 3Department of Hematology, University Medical Center Utrecht, 3584 CX Utrecht, The Netherlands; m.c.minnema@umcutrecht.nl

**Keywords:** primary central nervous system lymphoma, cerebrospinal fluid analysis, cytology, flow cytometry, diagnosis, meta-analysis

## Abstract

Primary central nervous system lymphoma (PCNSL) is a rare type of brain cancer that can often be treated successfully, if diagnosed early. PCNSL is often diagnosed through brain biopsy, which is invasive and risky. A safer and easier option is to test the cerebrospinal fluid (CSF), using techniques like cytology and flow cytometry. However, it is unclear how often these CSF tests are able to detect this disease. We reviewed studies from the scientific literature to find out how effective current CSF methods are in diagnosing PCNSL. These tests gave a positive result in about 18% of patients, but only 8% of all diagnoses were made using CSF testing. This suggests that while current CSF analysis is underused, it has the potential to play a bigger role in safely diagnosing PCNSL, especially with newer testing methods in development.

## 1. Introduction

Primary central nervous system lymphoma (PCNSL) is a rare form of aggressive, extra-nodal non-Hodgkin lymphoma, accounting for about 2% of all primary central nervous system (CNS) tumors [1]. It can involve the brain, spinal cord, leptomeninges, and the vitreoretinal compartment. Histopathological and immunohistochemical examination of brain biopsy material is the gold standard for diagnosing PCNSL, usually requiring stereotactic biopsy [2]. This neurosurgical procedure carries a risk of long-term morbidity, as well as a mortality rate between 0.8% and 2.2% [3,4]. Sampling error occurs in about 10% of cases [5]. Biopsy may be unfeasible due to lesion location, size, or comorbidities. Even when feasible, it remains burdensome and costly. Altogether, there is a clinical need for—ideally less expensive and non-invasive—alternatives for brain biopsy.

Analyses of cerebrospinal fluid (CSF) using cytomorphology or flow cytometry are such alternative diagnostic methods [6]. CSF cytomorphology can provide definite diagnosis by recognition of abnormal lymphoid cells [7], while flow cytometry has further enhanced the diagnostic value of CSF analysis in CNS lymphoma [8,9,10]. Recent guidelines by the European Association of Neuro-Oncology (EANO) and the European Hematology Association (EHA)/European Society for Medical Oncology (ESMO) recommend CSF analysis only as diagnostic module if biopsy is not feasible, as biopsy favors a precise lymphoma classification and CSF analysis could delay diagnosis due to limited diagnostic reliability [11,12]. In these guidelines, it is advised to perform a lumbar puncture only for examination of potential leptomeningeal involvement of patients with a final diagnosis based on biopsy. Hence, there are different thoughts about the role of CSF analysis in PCNSL: as a diagnostic tool in every patient suspected of PCNSL, which may replace a diagnosis from biopsy; as a diagnostic tool solely when biopsy is not technically feasible; or as a tool to detect leptomeningeal involvement to plan treatment strategies after tissue diagnosis by biopsy.

There has been limited systematic investigation into its diagnostic value in patients suspected of PCNSL. In this systematic review, we provide a comprehensive overview of data concerning the value of current standard CSF diagnostics (cytology and flow cytometry) in patients with PCNSL. Data on the diagnostic accuracy of CSF analysis, particularly its specificity, are largely missing, consistent with findings from a previously reported meta-analysis examining CSF involvement in lymphoid neoplasms [13]. That review reported, across three studies, high specificity for both cytology and flow cytometry. Given this assumed high specificity, aligned with clinical practice, we focus on (a) the detection rate of cytology and flow cytometry in current practice for diagnosing PCNSL; and (b) the proportion of diagnoses based on CSF analysis. Through this, we aim to quantify the current clinical value of CSF analysis for the diagnosis of PCNSL.

Certain biomarkers in CSF have shown promising diagnostic value for PCNSL, particularly the myeloid differentiation factor 88 (MYD88) L265P mutation in cell-free DNA (MYD88-cfDNA) and interleukin-10 (IL10) [12]. However, these markers have not been incorporated into most local guidelines due to the limited evidence base. Consequently, we did not include these markers in this review of current CSF diagnostics for PCNSL. Furthermore, a comprehensive review of new biomarkers has been published previously [14].

We also conducted a time-shift analysis to investigate whether this reported data changes over time. Finally, we addressed whether a second or third lumbar puncture is of added diagnostic value for PCNSL diagnosis.

## 2. Materials and Methods

This systematic meta-analysis was conducted following the recommended guidelines of the Preferred Reporting Items for Systematic Reviews and Meta-Analyses (PRISMA) Statement [15]. Ethical committee approval was not required. The protocol for the systematic review of our study was not registered for PROSPERO.

### 2.1. Search Strategy and Study Selection

A literature search of the PubMed and Embase databases was conducted, with the search limited to articles published up to 6 August 2024. The End-Note X9 library was used to remove any duplicates. The search strategy utilized a combination of keywords related to PCNSL and CSF analysis. We added keywords related to treatment studies of PCNSL, since these studies may include information on the diagnostic process of PCNSL patients. The search strategy is outlined in Appendix A.

The screening process was conducted by one author (JvR), who reviewed the titles and abstracts of the retrieved records and evaluated them for eligibility. This was followed by a full-text analysis of potentially relevant studies to check if they met the inclusion criteria. This analysis was performed by two authors (JvR and PB), and any disagreements regarding study eligibility were resolved through discussion with other authors (TJS and PW). In addition, the reference lists of all the articles were checked by snowballing techniques to detect additional studies not found by the electronic search.

### 2.2. Eligibility Criteria

The objective was to collect articles that contained the predefined domain, determinant, and outcome. Domain: patients (age ≥ 18 years) diagnosed with PCNSL based on histopathological confirmation (biopsy or resection) OR diagnosis based on CSF analysis. Determinant: information regarding CSF analysis (including cytology and/or flow cytometry). Outcome: diagnosis of PCNSL.

We included studies that contained data on adult (age ≥ 18 years) patients who (a) had a final diagnosis of PCNSL, (b) had available data on whether CSF analysis was performed and what type, and (c) if CSF analysis was performed, had available final results.

Only original (observational) cohort studies and case studies with a minimum of five PCNSL patients were eligible for inclusion. Original studies only focusing on lymphoma localization outside the CNS, recurring PCNSL, immunocompromised patients, and/or primary vitreoretinal lymphoma (PVRL) were excluded.

### 2.3. Data Extraction and Assessment of Study Quality

The two reviewers (JvR and PB) independently extracted relevant data from the papers included after the full-text analysis and simultaneously conducted a risk-of-bias analysis of all studies. The full-text analysis included recording the following information, if available:Publishing details: database, first author, and year published.Study characteristics: location(s), study type (retrospective or prospective), single or multicenter data collection, and period of enrolment.PCNSL data: total number of patients with PCNSL and inclusion method (diagnosis).Cerebrospinal fluid evaluation: positive results per cytology assessment, positive results per flow cytometry assessment, positive results from either method of assessment, number of patients diagnosed based on CSF analysis (conducted in patients with suspected PCNSL based on clinical and/or radiological findings), mention of and, if available, results of a second and/or third lumbar puncture.

In most studies, information regarding the percentage of patients whose diagnosis of PCNSL was based on CSF analysis was lacking. Therefore, the corresponding authors of these articles were contacted by email for this information.

The risk of bias was investigated with the use of the Quality Assessment of Diagnostic Accuracy Studies 2 (QUADAS-2) guidelines to assess the quality of primary diagnostic accuracy within the selected studies [16]. Each article was categorized as having high or low risk across four domains: (1) patient selection, (2) index tests (1: CSF cytology, 2: CSF flow cytometry), (3) reference standard, and (4) flow and timing. An overall risk-of-bias score was determined as high if 50% or more of the domains were categorized as having a high risk of bias. The assessment of potential bias was carried out by two independent authors (JvR and PB). In case of any disagreements, they held discussions with two additional authors (TJS and PW) to reach a consensus. No studies were excluded from the analysis solely based on risk-of-bias assessment.

A final screening was conducted by authors JvR and PB to exclude overlap between studies that were reported by the same research groups in overlapping periods of enrolment. If a possible overlap of enrolled patients was unclear, the corresponding authors of these articles were contacted by email. If overlap remained unclear (e.g., no reaction of the corresponding author was obtained), the articles with older publication dates were excluded from the final review.

### 2.4. Statistical Analysis

Primary data was used in a descriptive manner to estimate the percentage of CSF examination performed, specifically for cytology and flow cytometry, and the percentage of patients diagnosed with PCNSL was estimated through CSF examination. The detection rate estimates of the diagnostic tests from the included studies were pooled to conduct a meta-analysis using a random-effects model with the DerSimonian and Laird variance estimator [17]. Heterogeneity was estimated with Cochran’s Q test, mainly the I2 and χ42 statistics.

A time-shift analysis was conducted using meta-logistic regression to investigate a possible association between positive CSF findings and diagnoses using positive CSF results over the study period. The results are illustrated in meta-regression scatter plots.

The analyses were conducted in the R statistical software bundle, version 4.3.0, with the package ‘meta’, version 6.2-1, and Microsoft Excel 365.

## 3. Results

### 3.1. Study Selection

As can be seen in the flowchart of screened articles (Figure 1), the initial search resulted in 8103 studies, of which 2673 were duplicates. A total of 3976 studies were excluded after title–abstract screening, after which 1454 studies were analyzed for full-text assessment. Fifteen of these articles could not be retrieved. After full-text screening, a total of 1297 articles were excluded for the following reasons: (1) no information on CSF analysis; (2) limited information on CSF analysis, primarily no information on the total number of patients who were examined by CSF analysis; (3) the study population was not solely composed of PCNSL patients or included fewer than five patients with PCNSL; (4) other reasons, such as the article not being an original study. Finally, two records were added by snowballing techniques.

### 3.2. Study Characteristics

We finally included 144 studies in this review. The characteristics of all the included studies, as mentioned in the methods, are given in Table 1. The time range of the pooled studies was from 1975 to 2024. There has been an increase in the number of included studies per decade: 22% of the studies were published between 1975 and 2000, 27% between 2000 and 2010, 29% between 2010 and 2020, and 22% between 2020 and 2024. There were 90 retrospective cohorts, 50 prospective cohorts, and 4 studies that used both retrospective and prospective data. Most studies were included from Asian (36%), European (32%), and North American (21%) countries.

The final studies comprised patients diagnosed with PCNSL, with sample sizes ranging from 6 to 1002 patients. In summary, 72.2% of patients (6855 patients of 9493 patients diagnosed with PCNSL) had CSF examination. In most studies, no information was available for results on cytology or flow cytometry in individual cases. For example, in the largest study included [18], CSF examination was reported to be positive in 123 out of 594 patients (20.7%), but without separate information on flow cytometry or cytological examination results. Almost all the included articles (138 out of 144 articles, 95.8%) reported information on cytology. Flow cytometry results were reported in only 14 out of 144 (9.7%) articles.

Only 50 (34.7%) articles of the included studies reported that the final diagnosis of PCNSL could be solely based on CSF analysis. This information could only be retrieved from 43 articles because 7 articles mentioned that a final diagnosis of PCNSL could be solely based on CSF analysis but did not provide further information. In these cohorts, the final diagnosis was based solely on CSF in 5.8% (160/2741) of patients. In most other cases, diagnosis was confirmed by histological examination of brain biopsy samples, or, in very few cases, diagnosis was based on vitreous examination.
cancers-17-02352-t001_Table 1Table 1Study characteristics of 144 included studies.StudyCountryStudy DesignData Collection Site Period YearsTotal Number of Patients with PCNSLCytology n/NFCM n/NCytology or FCM n/NDiagnosed Based on CSFInclusion MethodFeldheim 2024 [19]GermanyRS2008–20221482/656/32UNAHTatarczuch 2024 [20]AustraliaRM2009–2018190NANA25/113UHYi 2024 [21]KoreaPM2018–2020353/35NA3/35UHJanopaul-Naylor 2024 [22]USARS2002–20219518/60NA18/60UHBatchelor 2024 [23]USAPM2012–20171088/88NA8/88UHZhang 2024 [24]ChinaRS2003–202169NANA9/67UUSchorb 2024 [25]GermanyPM2017–2020573/46NA3/46UH or C or VChuang 2023 [26]TaiwanRS1995–202112411/116NA11/116UHBairey 2023 [27]IsraelRM2001–202022224/155NA24/1557/221H or C or VMa 2023 [28]ChinaPS-3321/33NA21/33UURozenblum 2023 [29]FrancePM2016–20215412/41NA12/414/41H or C or VZhong 2023 [30]ChinaRM2007–2016961/96NA1/961/96H or CLi 2023 [31]ChinaRS2012–202264NA3/273/27UHLin 2023 [32]ChinaRS2018–20197726/73NA26/73UH or CBazer 2023 [33]USARS2010–2020132/90/92/90/9HEbrahimi 2023 [34]IranRS2011–2018586/58NA6/58NAHWang 2023 [35]ChinaRSU101/101/101/10UHDas 2022 [36]IndiaRS2013–2019997/7210/7210/72UHLage 2022 [37]BrazilRS2000–2021479/47NA9/47UHFerreri 2022 [38]5 countriesPM2010–201421934/162NA34/162UHVs 2022 [39]IndiaRS2011–2020422/42NA2/42UHYoon 2022 [40]South KoreaRS2012–202110623/67NA23/67UHLuo 2022 [41]ChinaPS2011–2018350/35NA0/35NAHSun 2021 [42]ChinaRS2001–201859UU27/59UHLee 2021 [43]South KoreaRS2015–2019222/22NA2/22UHYamagishi 2021 [44]JapanRS2009–2019391/39NA1/39UHGupta 2021 [45]USAP/RM2010–20191597/1107/104U6/110H or CFerreri 2021 [46]ItalyPS2016–2020365/36NA5/36UHSeidel 2020 [47]GermanyRS2015–2019436/26NA6/26UHShao 2020 [48]ChinaRS2013–20176625/66NA25/663/66H or CHouillier 2020 [18]FranceRM2011–20161002UU123/59433/594H or CSethi 2019 [49]USARM2000–2016536/51NA6/51UHLin 2019 [50]TaiwanRS2002–20181335/74NA5/74UH or C or VHiemcke-Jiwa 2019 [51]NetherlandsP/RS2016–2018222/138/1910/212/9H or C or VRimelen 2019 [52]FranceRS2016–201895/96/96/94/9H or CMao 2019 [53]ChinaRS2004–2017910/31U0/310/31HNayyar 2019 [54]USARSU64UU14/41UHPatekar 2019 [55]IndiaRS2001–20179913/82NA13/82NAHBromberg 2019 [56]3 countriesPM2010–201619917/139NA17/13917/139H or CMizutani 2018 [57]JapanRS2012–201790/6NA0/6UHZorofchian 2018 [58]USARM2010–2017150/84/94/11UHIkeguchi 2018 [59]JapanRS2006–201680/8NA0/8UHHottenrott 2018 [60]GermanyRS2015–2017681/444/53UUH or CNam 2018 [61]South KoraPS1996–201515419/131NA19/131UHAhn 2017 [62]South KoraRS1998–2012774/60NA4/60UHPark 2017 [63]South KoreaRS2002–2012626/57NA6/57UHPuligundla 2017 [64]IndaRS2005–2016426/38NA6/38UHJung 2017 [65]South KoreaRM2001–20156210/36NA10/36UHFan 2017 [66]ChinaRS2007–201610011/54NA11/54UHCerqua 2016 [67]ItalyRSU286/21NA6/21UHZhang 2016 [68]ChinaRS2006–2011280/28NA0/28UHJang 2016 [69]South KoreaRS2003–2014819/53NA9/53UULiu 2015 [70]ChinaRS2010–2014184/18NA4/18UUOmuro 2015 [71]FrancePM2007–20109523/78NA23/78UH or C or VPulczynski 2015 [72]MultiPM2007–2010668/49NA8/49NAHSasagawa 2015 [73]JapanPS2008–2014151/15NA1/15NAHLiu 2015 [74]China/CanadaRM2012–201383/84/84/8UUOlivier 2014 [75]FrancePM2000–2005351/23NA1/23UUFerreri 2014 [76]ItalyPM2000–2004413/36NA3/360/36H or C or VTao 2013 [77]ChinaRS2005–2009170/15NA0/15UHHe 2013 [78]ChinaRS1996–2011628/62NA8/62UHSalamoon 2013 [79]SyriaPS2006–2007408/40NA8/40UHRubenstein 2013 [80]USA/ItalyPMU385/374/207/38UUKorfel 2012 [81]GermanyPM2000–200936544/361NA44/3614/361H or CSasayama 2012 [82]JapanP/RS2004–2011310/22NA0/22UUWieduwilt 2012 [83]USAPS2001–2006316/26NA6/26UUGerard 2011 [84]CanadaPS1997–2006236/21NA6/212/21H or C or VLaack 2011 [85]USAPM1995–2000361/13NA1/13UH or C or VOmuro 2011 [86]FranceRM1994–2003646/55NA6/555/55H or C or VPasricha 2011 [87]IndiaRS1997–2009661/48NA1/48NAHFerreri 2011 [88]Italy/SwissPM2008203/14NA3/14UH or C or VSchoers 2010 [10]GermanyPU2008–2010233/235/236/23UHPels 2010 [89]GermanyPM1995–2001887/78NA7/78UUHohaus 2009 [90]ItalyRS1995–2004416/30NA6/302/30H or CAgarwal 2009 [91]IndiaRS2003–2006260/25NA0/25UHFerreri 2009 [92]MultiplePM2004–2007794/69NA4/69UH or C or VAngelov 2009 [93]USARM1982–200514911/121NA11/121UH or C or VIllerhaus 2009 [94]GermanyPM1998–2004292/11NA2/11UHKiewe 2008 [95]GermanyRS1994–2005726/34NA6/342/34H or C or VHaldorsen 2007 [96]NorwayRM1989–2003985/41NA5/412/41H or CYamanaka 2007 [97]JapanPS2003–2006112/11NA2/112/11H or CSilvani 2007 [98]ItalyPSU380/38NA0/38UHQuek 2006 [99]SingaporeRS1990–2005375/23NA5/23UUKawamura 2006 [100]JapanRM1995–1999468/40NA8/40UHOmuro 2005 [101]USA MSKCC RS1985–200018356/156NA56/156UUBrevet 2005 [102]FrancePS1998–200261/6NA1/6UHYamanaka 2005 [103]JapanPM1996–2003329/32NA9/324/32H or CHodson 2005 [104]UKRS1995–2003553/21NA3/21UHKorfel 2005 [105]GermanyPM1998–2000568/45NA8/453/45H or CDubuisson 2004 [106]BelgiumRS1987–2002326/11NA6/111/11H or CBessell 2004 [107]Spain/UKP/RM1986–2001724/36NA4/36NAHCaroli 2004 [108]ItalyRS1977–1997224/5NA4/5UHPoortmans 2003 [109]EuropePM1997–2002527/43NA7/43UH or CAbrey 2003 [110]USA/CanadaPMU283/28NA3/28UHIshikawa 2003 [111]JapanRS1981–1999339/33NA9/334/33H or CDabaja 2003 [112]USAPS1994–1996121/12NA1/12UHChoi 2003 [113]South KoreaRS1995–2001427/39NA7/39UHCheng 2003 [114]CanadaRS1998–200270/7NA0/7UH or VBatchelor 2003 [115]USAPM1998–1999253/14NA3/141/14H or CBraaten 2003 [116]USARS1988–2001333/11NA3/11UH or C or VFerreri 2003 [117]MRM1980–199937838/241NA38/24114/241H or C or VDeAngelis 2002 [118]MPMU9817/81NA17/81UH or C or VDepil 2002 [119]FranceRS1994–2000344/29NA4/29UUCalderoni 2002 [120]SwitzerlandRSU140/6NA0/6UHShibata 2002 [121]JapanRM1998222/10NA2/10UHGoldkuhl 2002 [122]Nordic countriesPM1997–1999305/30NA5/301/30H or C or VGleissner 2002 [123]GermanyRM1998–2001766/74NA6/74UUHerrlinger 2001 [124]GermanyRS1991–1997287/20NA7/201/20H or C or VMead 2000 [125]UKPM1988–1995534/34NA4/34UHZylber-Katz 2000 [126]IsraelPSU121/12NA1/12UHO’Brien 2000 [127]Australia/New ZealandPM1991–1997463/42NA3/42UHNg 2000 [128]AustraliaRS1995–1998102/6NA2/6UUWu 1999 [129]South KoreaRS1981–1997403/25NA3/25UHHiraga 1999 [130]JapanPS1992–1998295/23NA5/23UHGuha-Thakurta 1999 [131]USARS1993–1998312/242/133/241/24H or C or VBoiardi 1999 [132]ItalyPS1989–1994282/28NA2/28UHSandor 1998 [133]USAPSU147/14NA7/14UHCorry 1998 [134]AustraliaRS1982–19946210/34NA10/34UHCheng 1998 [135]TaiwanPS1991–199782/8NA2/8UHBrada 1998 [136]UKPS1986–1996318/26NA8/263/26H or C or VBlay 1998 [137]FranceRM1993–199522625/157NA25/1575/157H or CLaperriere 1997 [138]CanadaRS1979–1988491/17NA1/17UHGlass 1996 [139]UPU1990–1993182/14NA2/142/14H or C or VSchultz 1996 [140]USA/CanadaPM1988–1992526/45NA6/45UHSchaller 1996 [141]USARSU272/11NA2/11UHKrogh-Jensen1995 [142]DenmarkRM1983–1994482/5NA2/51/5H or CSarazin 1995 [143]FranceRS1989–1993222/22NA2/22UHBlay 1995 [144]France PS1983–1994252/25NA2/25UHGrangier 1994 [145]SwitzerlandRS1974–1990272/15NA2/15UHMiller 1994 [146]USARS1958–19891047/49NA7/49UHSelch 1994 [147]USARS1977–1992245/22NA5/224/22H or CGlass 1994 [148] USAPS1983–1990258/24NA8/245/24H or CHayakawa 1994 [149]JapanRM1970–19881709/42NA9/42UH or CLiang 1993 [150]UPU1988–199193/9NA3/90/9HFusejima 1992 [151]JapanRS1976–19893211/30NA11/300/30HRemick 1990 [152]USARS1980–1989134/9NA4/93/9H or CMichalski 1990 [153]USARS1966–1988364/22NA4/221/22H or CBrada 1990 [154]UKRS1963–1988355/16NA5/163/16H or CSocie 1990 [155]FranceRM1979–1987354/13NA4/130HGrote 1989 [156]USARS1974–1986122/11NA2/112/11H or CPollack 1989 [157]USARS1976–1986275/10NA5/100/10HVakili 1986 [158]USARS1964–1982261/14NA1/140/14HBogdahn 1986 [159]GermanyRSU107/8NA7/85/8H or CJellinger 1975 [160]Austria/HungaryRM1955–19756811/40NA11/40UHAbbreviations: PCNSL = primary central nervous system lymphoma; FCM = flow cytometry; CSF = cerebral spinal fluid; U = unknown; NA = not applicable; R = retrospective; P = prospective; S = single center; M = multicenter; n/N = positive cases/total tested; H = histopathological; C = CSF; V = vitreous.


### 3.3. Risk of Bias

A summary of the risk-of-bias analysis is provided in Appendix A. The percentages of articles with a high risk of bias for the four domains are as follows: (1) 64.6% for patient selection; (2) 77.5% for index tests (combining cytology and flow cytometry); (3) 41.7% for reference standard; and (4) 50.0% for flow and timing. Overall, 75.7% of the studies (109/144) were classified as having a high risk of bias. The percentages of articles with a high risk of bias differed between prospective cohorts (64.0%) and retrospective cohorts (78.9%), but this difference was not statistically significant (*p* = 0.07, Fisher’s exact test).

### 3.4. Meta-Analysis

According to the random-effects model in the forest plot of the meta-analysis, the overall pooled detection rate of positive CSF was 18% (95% CI: 16–20%; I^2^ = 67%) (Appendix A). The overall pooled detection rate for positive cytology was 17% (95% CI: 15–19%; I^2^ = 65%), while for flow cytometry, it was 20% (95% CI: 13–30%; I^2^ = 67%) (Appendix A). Meta-analysis revealed that the proportion of diagnoses based solely on CSF analysis was 8% (95% CI: 6–11%; I^2^ = 60%) (Figure 2).

### 3.5. Time-Shift Analysis

A meta-logistic regression was plotted, expressing the outcome of positive CSF results as a probability percentage over the period of the study (Appendix A). Similarly, a regression was conducted to define the probability of PCNSL diagnosis using positive CSF results (Appendix A). As can be seen in Appendix A, the values of the logistic regression coefficients were found to be −0.007 for a positive CSF result (95% CI: −0.001–−0.013; *p* = 0.029) and −0.023 for diagnosis based on the CSF result (95% CI: −0.007–−0.038; *p* = 0.004) (Appendix A). This indicates that over the included time period, the probability of a positive CSF result was 0.7% lower than that of a negative result. For diagnoses based on CSF results, this probability was 2.3% lower.

### 3.6. Repeated Lumbar Punctures

Only 2 out of 144 articles contained information regarding repeated CSF analysis [45,51]: Gupta et al. [45] reported 110 patients undergoing CSF analysis, with 13 patients (11.8%) receiving a second lumbar puncture (zero positive results for cytology or flow cytometry), 5 patients receiving a third lumbar puncture (one positive result, both on cytology and flow cytometry), and 1 patient receiving a fourth and fifth lumbar puncture (both without positive results). All CSF samples were taken prior to diagnosis. Hiemcke-Jiwa et al. [51] reported 22 patients undergoing CSF analysis, with five patients (22.7%) receiving a second lumbar puncture, of which three showed a positive result. For all three patients with a positive second result, the first lumbar puncture already showed a positive result. Hence, there was no added value. The reason for a second lumbar puncture was not stated; it was only noted that all CSF analyses were diagnostic.

## 4. Discussion

We performed a systematic review and meta-analysis of empirical studies looking into the value of standard CSF diagnostics (cytology and flow cytometry) for the diagnosis of PCNSL. These data serve to summarize the current ‘status quo’ of CSF examination in this context: current potential of well-established techniques, as well as actual use. Only 5.8% of PCNSL patients were diagnosed by CSF analysis, though 17.2% of the tested patients showed positive CSF results. Meta-analysis showed a pooled detection rate of 8% for diagnosis and 18% for positive CSF results. The discrepancy between the detection rate of positive CSF results and the percentage of diagnoses made using these results shows the gap in the potential of routine CSF analysis. Time-shift analysis indicates a decreasing probability of positive CSF results over time. Data on repeated lumbar punctures are scarce and were reported in only 2 of 144 articles. The percentage of positive CSF results (17.2%) in our systematic review is slightly higher compared to one other published systematic review looking into this subject by Morell et al. [161], with a positive CSF result in 14.9% of patients. The percentage of patients diagnosed by CSF in this systematic review was 3.1%, compared to 5.8% in our review. This systematic review included fewer studies (21 studies), excluding studies containing fewer than 20 patients.

In alignment with this systematic review, the review by Morell et al. also highlights the significant gap in the potential of CSF analysis but does not provide a possible explanation for it. To our knowledge, no study has explored the untapped diagnostic potential of CSF analysis. Several potential explanations can be offered: (i) The most significant reason is that brain biopsy with histological confirmation remains the clinical standard in most countries, with CSF examination typically reserved for detecting potential leptomeningeal involvement, following a final diagnosis based on biopsy, as outlined in the previously mentioned EANO and EHA-ESMO guidelines [11,12]. (ii) As a result of not using CSF examination as a diagnostic tool, some clinical centers either doubt its diagnostic value or do not utilize it. (iii) There is a potential delay in diagnosing PCNSL via CSF analysis, given its assumed low diagnostic yield [12]. (iv) Specific CSF tests, such as flow cytometry, may not be available. (v) A lumbar puncture may be omitted for clinical reasons, such as rapid deterioration of the clinical condition or the presence of space-occupying lesions that pose a risk of cerebral herniation. Results from our meta-analysis underscore the diagnostic potential of CSF analysis and the untapped value it holds. Strategies to overcome the practical and intellectual hurdles (mentioned above) for optimal use of CSF analysis are needed.

This systematic review shows increasing academic output on PCNSL diagnosis using CSF analysis techniques. Despite the introduction of flow cytometry around 1968, we did not find any increase in CSF-based diagnosis over time. In fact, the available studies show a minimal but significant decrease over time, which may be the result of the study types included over time. Also, the diagnostic yield of CSF analysis may decrease over time due to the increasing rate of neuroimaging: if the number of patients with an MRI-based suspicion of PCNSL and subsequent CSF investigation increases, while the number of patients with an ultimate diagnosis of PCNSL remains stable, then the diagnostic yield (detection rate) will decrease. In contrast to previous studies and reviews, flow cytometry did not significantly enhance the CSF-based diagnostic rate over time or result in a higher positive detection rate. Most of the previous studies looking into flow cytometry report on a small case series of unselected patients with hematological malignancies and only a small number of patients diagnosed with PCNSL.

Data regarding the use and yield of repeated lumbar punctures in the diagnosis of CSF is largely lacking, with only two articles stating this information [45,51]. On the one hand, Gupta et al. [45] reported 20 repeated lumbar punctures, of which only 1 (5%) showed a positive result. On the other hand, Bromberg et al. [9] provided data on repeated lumbar punctures in CNS hematologic malignancies, with 12 out of 60 patients (20%) showing a positive result after a first negative result. Since these patients are mainly diagnosed with systemic lymphoma (with screening of potential secondary CNS involvement), the added value of repeated lumbar punctures in the diagnosis of PCNSL remains unclear and cannot be endorsed or discouraged currently.

A major strength of this study is the rigorous search across multiple databases, analyzing over 5430 articles, with 144 papers eligible for the final review. We also provided a time-shift analysis and stated information regarding repeated CSF analysis.

This study has several limitations, both inherent to the meta-analysis and related to the interpretation of the results. The main limitation is that most studies included only patients with histologically confirmed PCNSL, possibly excluding those diagnosed by CSF alone, causing selection bias and underreporting of CSF-based diagnoses. Determining CSF involvement after the final diagnosis of PCNSL could lead to an overestimation of positive results (referral bias): a minimal atypical lymphoid cell in the CSF could be mistakenly interpreted as CSF involvement of PCNSL when considered in light of the final diagnosis. Another limitation is the lack of information on interobserver agreement and methodological heterogeneity in cytology and flow cytometry, leading to a high risk of bias [13]. For cytomorphology, in particular, distinguishing lymphoma cells from other pathological cells and/or normal leukocytes can be challenging, making interobserver variability likely.

Balmaceda et al. [162] were one of the first to propose CSF investigation following suspicions of PCNSL in 1995. They managed to diagnose 14.5% of patients with PCNSL only through CSF analysis, bypassing the need for brain biopsy. Balmaceda et al. even claim that this could be an underestimation, since a percentage of patients were treated with corticosteroids. In our study, we did not investigate the influence of corticosteroids prior to diagnosis on CSF analysis. This difference in diagnostic value of CSF analysis between our meta-analysis and the study of Balmaceda et al. illustrates that the diagnostic potential of CSF analysis may be increased by systematic application of CSF studies in clinical practice (as was performed by Balmaceda et al.). Despite the results of Balmaceda et al., nearly three decades later, we see a lack of systematic application of CSF analysis as a primary technique for PCNSL diagnosis.

Current international guidelines recommend CSF analysis only when a biopsy is not possible or to detect leptomeningeal involvement. CSF analysis has not yet been standardized as a diagnostic modality. Biopsy remains the clinical standard despite its risks and costs. Consequently, a more systematic use of CSF studies may provide patients with a low-risk, minimally invasive, and rapid diagnostic alternative. This is only feasible if clinical guidelines recognize the possibility of basing PCNSL diagnosis solely on CSF results. A potential limitation of CSF analysis is its inability to provide a definitive histological diagnosis or to classify lymphoma subtypes. Contrary to the EANO and EHA-ESMO recommendations, we believe that in a clinical setting, when a space-occupying lesion is identified with radiological suspicion of PCNSL, CSF analysis using cytology and flow cytometry can suffice—if positive—for diagnosing PCNSL. Moreover, CSF analysis should be considered for every patient suspected of having PCNSL, not only in cases where a biopsy is contraindicated, poses a high risk, or the patient is too weak to undergo the procedure. The currently demonstrated potential yield of CSF analysis for PCNSL, with a pooled detection rate of 18%, should be a key consideration for clinicians who still assume its diagnostic value is low. Recent research has identified novel biomarkers in CSF (microRNA, cell-free DNA, chemokines, cytokines) that may improve the diagnostic value of CSF in the upcoming years [14,163]. In particular, the use of MYD88-cfDNA seems promising, as this marker represents a lymphoma-specific mutation, which may carry a low prior risk of non-specific or false-positive findings. An overview of the advantages and disadvantages of these new techniques, alongside cytology and flow cytometry, is provided in Table 2. Preliminary findings of its diagnostic value are promising, but validation in prospective, adequately designed studies is needed [163].

## 5. Conclusions

In this systematic review of the current value of CSF analysis for the diagnosis of PCNSL in clinical practice, the most important finding is the gap between the overall detection rate of positive CSF outcomes and the proportion of diagnoses made using positive results. This shows that CSF analysis, a minimally invasive method to diagnose PCNSL, is underutilized in clinical practice. The findings of this systematic review should be interpreted with caution, as a significant number of the included studies exhibited a high risk of bias.

Recent research identified new biomarkers in CSF that could significantly improve its diagnostic utility by enhancing the sensitivity and specificity of CSF analysis for the diagnosis of PCNSL. Further research is encouraged to investigate the unexploited potential of current and new methods of CSF analysis in PCNSL. This may benefit patients through a more rapid diagnosis, with low procedure-related risks, resulting in the timely initiation of appropriate and potentially life-saving treatment.

## Figures and Tables

**Figure 1 cancers-17-02352-f001:**
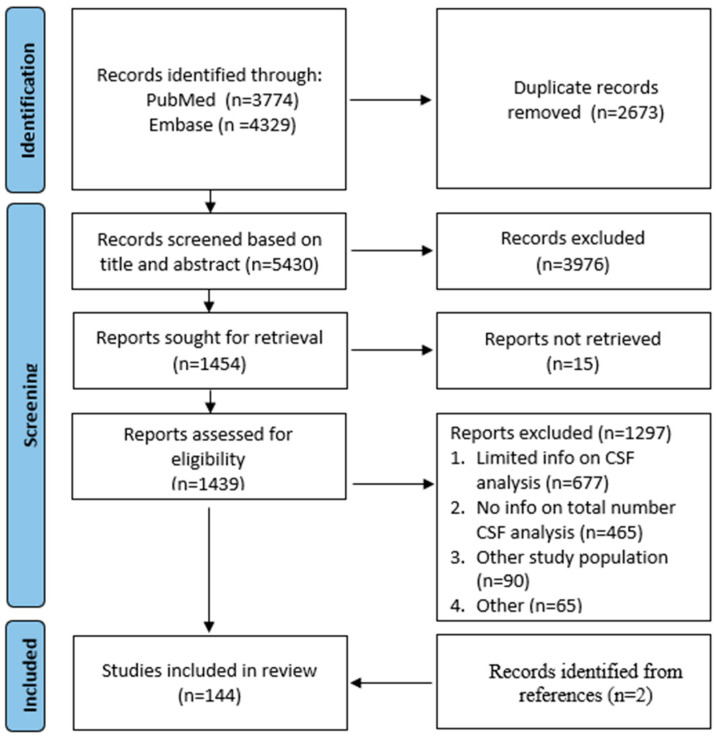
Flowchart illustrating study selection based on the PRISMA model.

**Figure 2 cancers-17-02352-f002:**
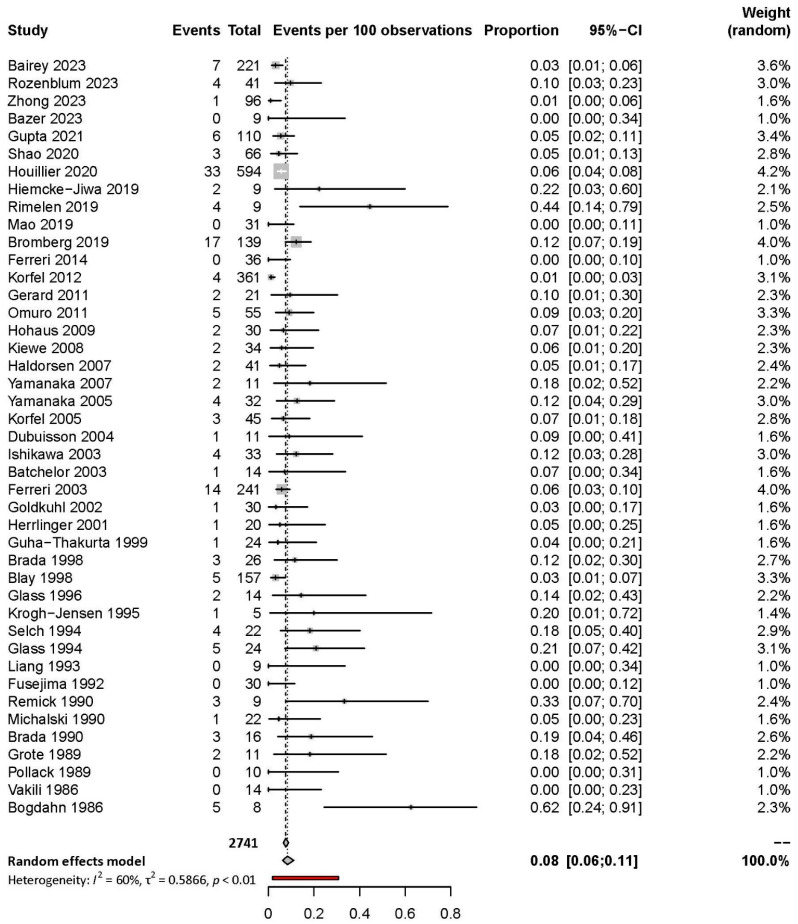
Forest plot of the proportion of diagnoses based on CSF analysis [18,27,29,30,33,45,48,51,52,53,56,76,81,84,86,90,95,96,97,103,105,106,111,115,117,122,124,131,136,137,139,142,147,148,150,151,152,153,154,156,157,158,159].

**Table 2 cancers-17-02352-t002:** Most commonly used diagnostic techniques in CSF for PCNSL: advantages and disadvantages.

Technique	Advantages	Disadvantages
Cytology/flow cytometry	Widely available	Limited sensitivity (17 to 20% detection rate)
	Assumed high specificity [13]	
	Enables basic immunophenotype classification	
Cell-free DNA analysis (MYD88-cfDNA)	Relatively high sensitivity (52 to 92%) [44,164]	Limited clinical validation
	High specificity (99%) [46]	Requires specialized equipment
Chemokines/cytokines (IL10/CXCL13)	High sensitivity (64 to 97%) [80,165]	Limited clinical validation
		Low precision can cause false positives, e.g., in neuroborreliosis [166]
		Cannot classify immunophenotype

## Data Availability

All data, as extracted from the original reports, can be found in the published tables and text.

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
