# Peer review of "The Diagnostic Yield of Cerebrospinal Fluid Analysis for the Diagnosis of Primary Central Nervous System Lymphoma: A Systematic Review"

_cancers, 2025, doi:10.3390/cancers17142352_

Round 1
Reviewer 1 Report
Comments and Suggestions for Authors
This is a thoughtful and well-constructed meta-analysis with a simple well stated goal. It is well written and makes its point regarding the under unitarization of CSF analysis in primary CNS lymphomas. I see no methodological issues and while the topic involves a relatively rare set of lymphomas, it has a relevant and important message. My only issue with the paper as written is not technical, but I find it surprising that more accurate and sensitive CSF based techniques have not been developed. The authors give several nice examples of molecular diagnostic techniques such as MYD88 PCR testing in Bing Neal syndrome with Waldenstrom’s Macroglobulinemia, but I think the paper would benefit from a more robust discussion of what can be done to improve the situation as biopsies in PCNSL, while the gold standard, should clearly be avoided if at all possible. Otherwise, this represents a nice addition to the literature and one that will hopefully spur more diagnostic research and development as well is change clinical practice.
Reviewer 2 Report
Comments and Suggestions for Authors
This work deals about value and use of lumbar puncture in PCNSL diagnosis. The paper is interesting as the argument is now in the interest of the general audience. Find a less invasive method than biopsy to diagnose PCNSL will represent a positive upgrade in the clinical practice. Authors made a great work analyzing a large number of papers and include in the study is 144 for a total of 9943 patients included. The literature review ranged from 1975 to 2024 covering a large period with obvious methodological changes in CSF examination. The manuscript contains some interesting data and some novelties as it can reports the percentages of execution and usefulness of the PL. Percentage of execution percentage of diagnostic power and of repetition of PL. The paper has a subsidiary role. However the scientific value if this data is not so clear. The study should be addressed to sustain new important studies. The role of PL should only be clarified with prospective and will designed studies. This should be underlined in the text. I consider the introduction should be shortened a little. Presentation of the bias risk analysis is correct. Methods are clearly presented and results are detailed offering a picture about the clinical use of CSF examination in PCNLS diagnosis. I also recommend a discussion reduction with more focus on future studies. Bibliography only contains 23 element and need to be considered as the PRISMA Diagram include 144 article in the meta-analysis and review. This should be better explained.
Comments on the Quality of English LanguageThe quality of English is good, the paper is readable and understandable. I only suggest to reduce the use of the adverb solely that is quite redundant.
Reviewer 3 Report
Comments and Suggestions for Authors
This is a comprehensive review about CSF analyses (cytomorphology and flow cytometry) for the diagnosis of PCNSL. I have some comments about it.
#Introduction, Discussion, or somewhere else
It may be good/reasonable for this paper to devote to cytomorphology and flow cytometry of CSF for the diagnosis of PCNSL. But as the authors also address, some other diagnostic modalities like MYO88, interleukin-10, or others are now also available at hand. It is not necessary to delineate the latter (MYO88, interleukin-10, or others), but it would be wonderful to have a comprehensive table about the PROS and CONS regarding the issue of ‘cytomorphology/flow cytometry vs MYO88, interleukin-10, or others’ for the CSF diagnosis of PCNSL in this review. Otherwise, the readers cannot understand why the authors had to stick to cytomorphology and flow cytometry at one end, albeit having MYO88, interleukin-10, or others at the other end. Just to cite a paper (ref 14) is not enough for good understanding of the readers.
#’3.2 Study characteristics’ or Table S2
-It is boring to read this section (3.2 Study characteristics). To look at this Table S2 as much as possible is more appealing than just reading this section. Probably, no one would dare to read it carefully. How about incorporating this table as an official table (not supplementary) into the MS, and incorporating its refs into the ‘References’ of the MS?
-Table S2: It is rather unclear what n/N indicates (Cytology ‘n/N’ FCM ‘n/N’ Cytology or FCM ‘n/N’). At least, I could not understand it. It would be better to explain this much better somewhere in the MS.
#’3.3 Risk of bias’ or Table S3
This is the same as above. To look at this Table S3 as much as possible is more appealing than just reading this section. How about incorporating this table as an official table (not supplementary) into the MS?
#Throughout the Results, tables/figures, supplementary tables/figures
This is just what I wonder as a reader of the MS. It oftentimes says ‘positive CSF results’, ‘diagnoses based on CSF analysis,’ or so. I guess these terms indicate that it is derived from both of cytomorphology and flow cytometry, thus ‘positive’ or ‘diagnoses based on CSF’, right? But I just wonder whether there is a case (ref) where it is derived from either of them, but not from both of them. To tell whether it is from both or either of them for each case (ref) is impossible, or after all, is redundant?
#Some questionable points are noted about English grammar or syntax. Please check the MS again.
-line 15: a rare type of brain cancer that >>> a rare type of brain cancers that (?)
-line 89-90: include these markers this review of current CSF diagnostics >>> include these markers in this review as current CSF diagnostics (?)
-line 92: also conduct >>> also conducted
-line 92-93: whether this reported data changes over time (???)
-line 233: =60%) Figure 2) >>> =60%) (Figure 2) (?)
-etc. etc. etc.
Round 2
Reviewer 3 Report
Comments and Suggestions for Authors
Now that the MS has been revised as was expected, it is one of good reviews about CSF analysis for diagnosing PCNSL.